# What Patterns in Online Classified Puppy Advertisements Can Tell Us about the Current UK Puppy Trade

**DOI:** 10.3390/ani13101682

**Published:** 2023-05-18

**Authors:** Katharine Eloise Ross, Fritha Langford, Dominic Pearce, Kirsten M. McMillan

**Affiliations:** 1Royal (Dick) School of Veterinary Studies, University of Edinburgh, Edinburgh EH25 9RG, UK; 2Animal and Veterinary Sciences, Scotland’s Rural College (SRUC), Edinburgh EH25 9RG, UK; fritha.langford@newcastle.ac.uk; 3School of Natural and Environmental Science, Newcastle University, Newcastle upon Tyne NE1 7RU, UK; 4Fios Genomics BioQuarter, 13 Little France Rd., Edinburgh EH16 4UX, UK; dominic.pearce@fiosgenomics.com; 5Dogs Trust, London EC1V 7RQ, UK; kirsten.mcmillan@dogstrust.org.uk

**Keywords:** puppy trade, data science, animal welfare, canine welfare, conformational disorder

## Abstract

**Simple Summary:**

Over the last ten years, the UK (United Kingdom) puppy trade has moved almost entirely online. Breed demand and impulse buying have made puppies lucrative commodities in an industry driven by profitability. To compensate, production levels have risen, feasibly fuelled by unethical breeding, poor husbandry/handling practice, and reduced overheads. As a result, breeding stock and puppies may have suffered potentially fatal physiological and long-term psychological issues. Surveys of veterinary professionals report an increase in sick puppies purchased online, whilst the British Small Animal Veterinary Association (BSAVA) reports an epidemic of canine inherited disease. The rapid growth of the puppy trade leaves significant gaps in regulation, and, to date, the nature of online trade remains difficult to quantify; however, a wealth of information can be gleaned from online advertisements, voluntarily posted to the public domain. This includes litter size, breed, seller location, activity, and more. In this study, a buoyant, consumer-driven market was revealed over a two-year period, in which the top 20 most advertised breeds were in accordance with the Kennel Club (KC) registered breed statistics, signifying a significant overlap with real-world data. Of all puppies advertised online, 66% were of 20 breeds, and 46.8% of puppies were listed as breeds linked to conformational disorders. Regional and seasonal fluctuations in price were evident, suggesting a market influenced by consumer trends. The highest number of advertisements per capita were located within Wales, the reported ‘puppy farm capital’ of the UK. Mean price for all breeds was GBP 927.14, increasing by over GBP 150 per individual, over the two-year duration of the study.

**Abstract:**

The UK online puppy trade has rapidly outgrown the current legislation, aided by the anonymity provided by classified advertisement platforms. In an effort to meet increased demand, some unregulated and regulated breeders may have employed practices that negatively impact canine welfare. A paucity of up-to-date empirical data, necessary to characterise the scale and nature of this industry, makes intervention challenging. This study quantifies the online puppy trade via web-scraped online classified advertisements, providing empirical data that reveal market trends, along with spatial and temporal patterns. A total of 17,389 unique dog advertisements were collated and analysed over a 2-year period (1 June 2018 to 31 May 2020). The second year included the COVID-19 Lockdown (23 March 2020 to 31 May 2020). Statistical comparisons between dependent and independent variables were performed by linear regression. In the case of a single continuous variable, a one sample *t*-test was used. Of these advertisements, 57.2% were sourced from a pet-specific classified advertisement website (Pets4Homes, n = 9948), and the remaining 42.8% from two general classified advertisement websites (Gumtree, n = 7149, 41.1%; Preloved, n = 292, 1.7%, respectively). England exhibited the greatest number of advertisements (n = 10,493), followed by Wales (n = 1566), Scotland (n = 975), and Northern Ireland (NI; n = 344). Scaled for estimated human population density, Wales possessed as many advertisements per million inhabitants (489.4) as the other three combined (England = 186.4, Scotland = 177.3, and NI = 181.1). Across both years, 559 unique breeds were advertised, yet 66% of all advertisements focused on 20 breeds, and 48% advertisements focused on only 10 breeds. Regional breed popularity was suggested, with French Bulldog as the most advertised breed in England (7.3%), Scotland (6.8%), and Wales (6.8%), but Schnauzers were most popular within Northern Ireland (6.83%). Within the 559 unique breeds advertised, only 3.4% had links to conformational disorders CD); however, these breeds were among the most commonly advertised, totalling 46.9% of all ads. Across all regions, price density peaked between GBP 300 and GBP 1000, with Bulldogs presenting the greatest cost (mean = GBP 1461.38, SD = GBP 940.56), followed closely by French Bulldog (mean = GBP 1279.44, SD = GBP 664.76) and Cavapoo (mean = GBP 1064.56, SD = GBP 509.17). CD breeds were found to be GBP 208.07 more expensive, on average, than non-CD breeds. Our results represent a buoyant online market with regional and seasonal fluctuations in price, advertised breed frequency and total counts. This suggests a market influenced by consumer trends, with a particular focus on breed preference, despite links to illness/disease associated with conformation. Our findings highlight the value of utilising online classified advertisement data for long-term monitoring, in order to assist with evidence-based regulatory reform, impact measurement of targeted campaigns, and legislative enforcement.

## 1. Introduction

Dogs are the most popular pet in the UK, with 30% of UK households reportedly owning a dog in 2021 [1]. Underpinning this demand for dogs, is a puppy market whose profit-driven reality could be said to lay in stark contrast to the values on which the human–animal bond is founded. A 2016 report by the Royal Society for the Prevention of Cruelty to Animals (RSPCA) showcased the modern puppy industry as one driven by maximising profit, often at the expense of canine welfare [2]. Much of this trade supply is suggested to originate from large-scale, industrial-style operations, in which breeders may achieve low overheads by practising poor husbandry and deficient hygiene standards. These operations are colloquially dubbed ‘puppy farms’ [2]. However, issues of puppy welfare may also extend to regulated and small-scale breeders, by means of weak regulation and/or the practice of inbreeding (“line breeding”) to achieve pedigree breed standards. The popularity of certain breeds may influence breeders to favour popular breeds, including purebred dogs, bred to KC breeding standards or ‘designer’ cross-breeds (DCBs) in order to maximise profits and meet demand [2]. In many cases, desirable aesthetic features are the product of inbreeding and positive selection for increasingly extreme physical traits, without regard for associated health issues. There is much evidence to showcase inbreeding to achieve pedigree breed standard’s association with hereditary pathology and conformational disorders [3,4], including orthopaedic and joint disorders [3,5,6]; skin disease [3,7]; aural disease [3,8,9]; ocular disease [4,10]; and breathing difficulties, such as those resulting from Brachycephalic Obstructive Airway Syndrome (BOAS) [4,11,12,13]. Holland’s 2019 review of factors contributing to dog acquisition behaviour concluded that identifying underlying breed trends could enable stakeholders to respond to consumer behaviour more appropriately [14].

Conceivably, efforts to introduce legislation to protect the welfare of animals within the United Kingdom (UK), along with effective education, media attention, and intervention schemes, may have contributed to a supply deficit of puppies bred in the UK as higher standards may increase overheads and production lag [15]. This, in turn, is likely to have encouraged puppy importation and smuggling, with puppy farms operating outside of UK jurisdiction supplying large sections of the British and UK market [15,16,17]. Characteristics of these supply routes may include fraudulent advertising, third party sales, accelerated separation from the mother, early (or no) vaccination, the transport of heavily pregnant bitches, and poor transport conditions [18]. Lax border checking regimes and low visual inspection rates, paired with insufficient fines and lack of sentencing for those caught illegally importing puppies, have failed to deter criminals from seeking out this high-profit, low-risk trade [15]. Consequently, puppy smuggling has been linked to organized crime, with puppies being coined the ‘new narcotics’ [19].

Meanwhile, within the puppy population, UK veterinary practices are reporting an increased incidence of severe to fatal infections, some occurring within days of acquisition, alongside an increase in chronic conditions that are most likely inherited (e.g., respiratory disorder, skin fold dermatitis, otitis, and spinal disease) [20]. Furthermore, smuggled dogs are commonly un-vaccinated and pose a public health risk with regard to exotic/zoonotic disease transmission (e.g., Rabies, Leishmania) [16,18,21]. These factors, coupled with a naïve consumer market liable to indulge in fashionable breeds and impulse purchasing, has caused the puppy trade to rapidly outgrow even recent regulation [14,15,22]. Despite warnings of a ‘major dog welfare crisis’ on the horizon [23] the UK puppy trade remains a thriving industry, with an estimated annual sale of 0.8–1.3 million puppies [24].

The internet has been recognised as one of the main facilitators for irresponsible and illegal selling and buying behaviours [2,25,26]. The online puppy ‘product’ can be described idealistically, with unrelated and/or fabricated descriptors attached to advertisements [27]. The anonymity and the opportunity to reduce face-to-face contact makes online advertisement the ideal platform for sellers to encourage and prosper from impulse purchases, especially for those who do not wish their facilities to be scrutinized [15]. Several animal welfare organizations have recommended that buyers observe puppy and mother in their home environment, prior to purchase [28]. However, sellers have been reported to imitate reputable breeders by renting family homes to showcase stand in ‘mothers’ [18]. Sellers have also been reported to use aliases and/or burner phone numbers to sever contact post-sale [2].

Within the market economy for pets, online systems distort the market for puppies, by potentially providing an effective sales platform for those prioritizing profit over purpose [29]: unscrupulous breeders are being provided with a tool with which to financially undercut those breeding to high standards [29]. Furthermore, a lack of buyer screening and ease of access (e.g., quick turnover, direct delivery of puppies to buyers) can increase temptations to purchase [29]. Thus, this increased consumer convenience and immediate availability will run contrary to good breeding/sale practices. As such, there is a clear need for regulatory reform, legislative change, targeted campaigns, and/or enforcement to counteract this.

To date, there is a critical lack of up-to-date empirical data regarding the UK’s online puppy trade, with current statistics providing only partial estimates based on government, trading standards, local authority, NGO, and commercial/social enterprise reports [15]. These studies provide an imperative contextual insight regarding the nature and scope of the industry, e.g., the impact on animal welfare [30], environmental/public health concerns [15], and links to crime (within the context of substandard regulations and gaps in legislation) [19]; however, they do not reliably assess nor evaluate spatiotemporal patterns in the trade. Consequently, this study aimed to:(1)Reliably quantify spatial and temporal patterns within the UK online puppy trade;(2)Pilot an analytical pipeline, that may be used to develop proactive response strategies regarding canine welfare issues, and measure impact evaluation regarding human behavioural intervention campaigns and/or changes to legislation.

## 2. Materials and Methods

### 2.1. Data Sourcing

Data were collected over a 24-month period, via web-scraping three popular classified advertisement (ad) websites advertising puppies for sale in the UK, Gumtree, Preloved, and Pets4Homes. Web-scraping and data collation were carried out by Hindesight Ltd. (Registered 08216374, UK). The raw data received consisted of individual ads, the incorporating source website, the seller location, the advertisement title, the date posted, the unique advertisement ID, the unique seller ID, and the unique litter ID. In order to comply with the University of Edinburgh’s Human Ethical Review Committee (HERC) approval requirements and General Data Protection Regulation (GDPR) (EU 2016/679), unique human IDs were removed before the data were provided for analysis.

### 2.2. Data Processing

The raw data were processed to ensure a single analysis unit (i.e., an advert) was defined by the source website, advertisement title, date posted, unique advertisement ID, unique seller ID, and unique litter ID. The seller location, price and breed were further extracted, cleaned, and categorized (discussed below). Advertisements were categorised into one of two sampling phases of 365 days in length, Phase 1, 1 June 2018–31 May 2019 and Phase 2, 1 June 2019–31 May 2020 (see Table 1). Due to missing data in December of Phase 2, all December advertisements for both phases were excluded from the downstream analysis. Data cleaning included the removal of duplicate or ambiguous ads, i.e., those missing date posted, advertisement title, or seller location, or possessing identical or missing unique advertisement IDs. When comparing price across phases, prices were normalized as a proportion of median income for respective tax years [31,32]. Cleaning also included the removal of non-sale ads, e.g., ‘stud’ services. Source websites were checked for misspelling and assigned as one of three popular classified advertisement platforms. advertisements were assigned to a UK region (England/Scotland/Wales/Northern Ireland (NI)) based on their county information (sample in Appendix A). Breed names were extracted from advertisement titles by searching for the presence of a pre-defined term within a ‘breed dictionary’, made available by Dogs Trust Research Team (sample in Appendix B). This breed dictionary is based on The UK Kennel Club (KC) and/or Fédération Cynologique Internationale (FCI) recognised ‘purebred’ list, with some popular breed additions (e.g., Lurcher, Boerboel; refs. [33,34]). The dictionary included a range of breed misspellings and breed name alternatives, allowing for the classification of a diverse range of descriptions to a reduced vocabulary. For example, both ‘cockapoo’ and ‘cocker spaniel poodle’ were converted to ‘cocker spaniel × poodle’). Advertisements identified as including more than one breed were listed as cross breed, using the first breed specified as the cross, e.g., ‘cocker spaniel cross/type’. Advertisements unable to be assigned a breed were excluded from the analysis. Cleaned breeds were then classified into those reported to display breeding-related conformational disorders (CD) or not, as per Asher et al. [3] where the ‘not’ includes cross breeds, as there is as yet unpublished evidence of CD in such crosses. Advertisements were filtered to exclude extreme prices, i.e., >GBP 5000 or <GBP 200. This was to remove advertisements detailing pet accessories, adoptions, or spurious posts. A detailed description of the data cleaning pipeline is listed within Appendix C. In cases where effectiveness of data cleaning could not be evaluated programmatically, a random selection of 50 advertisements were manually checked to ensure data validity. In compliance with HERC approval conditions with regard to the use of personal or identifying data, examples of this cannot be presented within this paper; however, the unique set of terms populating excluded advertisements can be found in Appendix D.

### 2.3. Statistical Analysis

Statistical analyses were performed using the R programming language (version 4.0.2 (1 April 2021), ref. [35]). A comprehensive list of R packages used can be found in Appendix E. The manuscript and figures were composed in R markdown and rendered using XeLaTeX via Pandoc [35]. Statistical comparisons between dependent and independent variables were performed by linear regression, unless otherwise specified. In the case of a single continuous variable, such as a measure of the relative difference between phases, a one sample *t*-test was used. Pearson’s correlation coefficient (Pearson’s r) was calculated to determine the relationship between two continuous variables. In the case of non-normality, continuous data were log_2_ transformed. In the case of fold changes, this resulted in transformed values that were symmetrical about zero, aiding interpretation, wherein a doubling or a halving would, thus, be represented by the same coordinate distance (+1 and −1, respectively). For comparisons where data normality could not be reliably assumed, and log_2_ transformation was not appropriate, a permutation-based approach was used. Categorical similarity was assessed using the Jaccard similarity index, where an intersection (i.e., the overlap) is calculated as a proportion of its union (i.e., all unique values in both categories combined). To test whether a time-series was stationary, the Augmented Dickey–Fuller Test was applied. This test relies on a null hypothesis of being non-stationary. Where appropriate, locally estimated scatterplot smoothing (LOESS) was used for visualisation purposes.

## 3. Results

### 3.1. Data Suitability

To determine whether the data could reflect the offline puppy trade, similarity between the top 20 most advertised breeds online were assessed against the KC’s 2019 top 20 most registered breeds, with an intersection of 10 breeds apparent (Jaccard index = 0.33). To determine the significance of this, 20 random breeds were repeatedly randomly drawn from the data 10,000 times and the Jaccard index recalculated for each, essentially providing an estimated overlap that could be expected by chance. In no random draw was a Jaccard index achieved as high as that seen when comparing the actual observed data’s top 20 most commonly advertised breeds (*p* < 0.001), suggesting a high degree of similarity between the datasets. Our dataset was therefore considered a meaningful reflection of the KC dataset for popular breeds and suitable to be interpreted as a reflection of puppy sales more generally.

### 3.2. Descriptive Statistics

A total of 27,081 advertisements were collected over a 24-month period, from 1 June 2018–31 May 2020 and split into two, 365-day phases (see Table 1). The second 365-day period included a full COVID-19 Lockdown from 23 March 2020 to 31 May 2020, where movements of the general population in the UK were restricted. Following data cleaning (see Appendix C), 17,389 advertisements were available for analysis. Within the 17,389 ads, 57.21% were sourced from a pet-specific classified advertisement website (Pets4Homes, n = 9948), and the remaining 42.79% from two general classified advertisement websites (Gumtree, n = 7149, 41.11%; Preloved, n = 292, 1.68%, respectively).

Total advertisements per-day was plotted as a time-series, suggesting a non-stationary relationship between frequency and time (*p* = 0.38, Augmented Dickey–Fuller test), with peaks observed in April and May (Figure 1). It was observed that advertisement frequency decreased between Phase 1 and 2, demonstrating a negative log_2_ relative advertisement frequency (mean = −0.22, 95% CI = [−0.15, −0.29], *p* < 0.001).

England exhibited the greatest number of advertisements (n = 10,493), followed by Wales (n = 1566), Scotland (n = 975), and Northern Ireland (n = 344). Scaled for estimated human population density [31,32], Wales possessed as many advertisements per million inhabitants (489.38) as the other three regions combined (England = 186.38, Scotland = 177.27, and NI = 181.05).

Price density peaked between GBP 300 and GBP 1000 (Figure 2), with a mean value of GBP 927.14 (SD = 604.03). Additional peaks were observed around notable ‘milestone’ values, e.g., GBP 1250, GBP 1500, GBP 2000, GBP 2500, GBP 3000, etc. Very few advertisements (0.13%, n = 22) posted prices over GBP 4000.

Regional price differences were noted (*p* < 0.001, ANOVA), with English advertisements being the most expensive on average (mean = GBP 968.36, SD = GBP 614.9), then Wales (mean = GBP 952.5, SD = GBP 604.97), Scotland (mean = GBP 905.21, SD = GBP 532.8), and Northern Irish being the least expensive (mean = GBP 514.38, SD = GBP 345.88).

Amongst breeds individually responsible for ≥0.5% of all ads, the median price per breed was not significantly correlated with breed advertisement frequency (r = 0.1, 95% CI = [−0.22–0.4], *p* =0.54). Across all regions, bulldogs were the most expensive (mean = GBP 1461.38, SD = GBP 940.56), followed closely by French Bulldog (mean = GBP 1279.44, SD = GBP 664.76) and Cavapoo (mean = GBP 1064.56, SD = GBP 509.17) (See Appendix F for mean price of top 20 advertised breeds).

### 3.3. Breeds

Across both phases, 559 unique breeds were advertised, yet 66% of all advertisements focused on 20 breeds, and 48% of advertisements focused on only 10 breeds. The 10 most advertised breeds were French Bulldog (n = 1285, 7.39%), Chihuahua (n = 1026, 1026%), Cockerpoo (n = 1022, 5.88%), Labrador Retriever (n = 916, 5.27%), Cocker Spaniel (n = 856, 4.92%), Pug (n = 821, 4.72%), Dachshund (n = 803, 4.62%), Bulldog (n = 750, 4.31%), German Shepherd (n = 459, 2.64%), and Shih Tzu (n = 439, 2.52%) (See Appendix G for top 20 most advertised breeds by region).

Whilst overall breed rankings, i.e., the total number of ads, was observed to fluctuate between Phase I and Phase II (mean shift in rank = 5.58), the top 20 remained very consistent (mean shift in rank = 0). Amongst the 20 most popular breeds, Cavalier King Charles Spaniel was the breed that increased in rank the most between phases (+4, 20th–16th), followed by Dachshund (+3, 8th–5th), and Jack Russell Terriers (+2, 13th–11th). Conversely, for this same subset, the largest decrease in rank was −3, held by Pugs (5th–8th), Shih Tzus (9th–12th), and Yorkshire Terriers (15th–18th).

Ads for purebreds categorised as brachycephalic (Boxer, Bulldog, Shih Tzu, Pomeranian, Chihuahua, French Bulldog, Pug, and Cavalier King Charles Spaniel [4]) showed a decrease in advertised frequency of 8.37% between phases.

The most commonly advertised breeds varied between regions (Figure 3).

The advert frequency for the top 20 most commonly advertised breeds varied significantly between regions (χ^2^ = 197.33, df = 180, *p* = 0.18, χ^2^ test, Figure 3). The French Bulldog was the most commonly advertised breed in England (7.32%), Scotland (6.83%), and Wales (6.83%). In Northern Ireland, Schnauzers were most commonly advertised (6.83%).

### 3.4. Conformation Disorder

Within the 559 unique breeds advertised, only 3.4% had links to conformational disorders (CD); however, these breeds were among the most commonly advertised, totalling 46.85% of all ads. Amongst breeds individually responsible for ≥0.5% of all ads, CD breeds were observed to be advertised significantly more frequently than their non-CD counterparts (β = 298.64, SE = 96.74, *p* < 0.01, Figure 4b).

### 3.5. Price

As advertisement frequency was observed to vary by time and CD status, it remained to assess the relationship of these factors with price, in the hope they may provide insight into market supply.

Advert prices were summarized as the median per-breed and compared across phases, wherein median Phase 2 prices were calculated as a proportion of their Phase 1 equivalents and log_2_ transformed to ensure data symmetry around zero. For a specific breed, a value of 1 indicated a doubling, 2 a quadrupling, and −1 a halving in Phase 2 relative to Phase 1. For this analysis, price was normalized by median income for tax years 2018–2019 (Phase 1) and 2019–2020 (Phase 2) [31,32].

Contrary with advertisement frequency, log_2_ price fold change between Phases 1 and 2 was observed to increase (mean = 0.24, 95% CI = [0.16, 0.31], *p* < 0.001, One Sample *t*-test; Figure 5a). This trend was consistent for the top 20 most popular breeds, with all but French Bulldog prices increasing between phases (Figure 5b).

Price was then modelled as function of CD, phase, and their interaction (Figure 5c).

The estimated coefficients suggested that in Phase 1 the mean price for a non-CD breed was GBP 712.5, whilst a CD breed was found to be GBP 208.07 more expensive on average. In Phase 2, non-CD breeds were observed to be GBP 329.21 more expensive than their Phase 1 counterparts. However, this increase in price between phases was tempered in CD breeds, with a significant (*p* < 0.001) interaction between CD and Phase being evident, coincidental with an additional mean decrease in price of GBP −128.61. In essence, non-CD dog prices appeared to close the price gap on their CD counterparts in Phase 2.

## 4. Discussion

This study aimed to investigate patterns in the UK puppy trade by way of online classified advertisements, collected between 1 June 2018 and 31 May 2020 (Phase 1, 1 June 2018–31 May 2019 and Phase 2, 1 June 2019–31 May 2020). Here, we provide quantitative estimates regarding the online puppy trade, revealing spatiotemporal patterns within breed popularity and price, at both the UK and country-level scale. Web-scraped is free of many of the biases presented in other human-behaviour study data, such as that from surveys [36]. This supports our findings from this study in supporting the value of online classified advertisement data as a reliable source for long-term monitoring of the UK puppy trade.

Our first objective was to investigate the suitability of the data as a representative sample of the offline puppy trade. Herein, the overlap between the top 20 most advertised breeds online and top 20 KC registrations in 2019 was assessed. Ten breeds of each dataset were observed to overlap, a number significantly greater than would be expected by chance (*p* > 0.001). However, as the KC record registered ‘purebreds’ only, their data will include biases with regard to the overall UK dog population. The resulting knowledge gap could be potentially large, as the KC annually registers ~250,000 dogs, whilst PFMA has recorded an annual increase of 850,000–3.5 million dogs every year for the last 4 years [1]. This discrepancy may be, in part, due to the increasing popularity of ‘designer crossbreeds’ (DCBs) (the planned mating between distinct pure breeds to create new ‘designer crossbreeds’ with desirable physical and reported temperamental attributes and given a ‘catchy’ name) [37]. Consequently, the observed intersection between the two datasets would likely be greater if KC registrations included DCBs, which feature prominently in the top 20 breeds advertised online. Thus, we consider the similarity index a conservative estimate, and, as a result, believe that online classified advertisement data provide a meaningful reflection of UK puppy sales.

Our second objective was to provide quantitative spatiotemporal estimates of the UK puppy trade. Advertisement density revealed a buoyant market with seasonal fluctuations, displaying an apparent sharp increase in advertisement counts in April–May. Western companion dogs do not experience season-specific oestrus, as seen in wild dogs or street dog populations, though some evidence suggests breeders may induce oestrus to meet demand [38]; however, this practice is uncommon. Seasonal demand fluctuations are likely due to an increase in demand during school/summer holidays, when more time may be dedicated to a puppy (suggesting a consumer-driven market), but how the supply is meeting this demand is unknown.

Although the majority of advertisements were located in England, Wales proved the UK’s most prolific region for sales of puppies, when calculated per-capita. This could support the moniker ‘puppy farm capital’, given to Wales after a recent investigative report on both legal and illegal Welsh puppy farming industries [39]. In 2003, the Welsh Assembly introduced grants to encourage struggling farmers to diversify into dog breeding [40] in order to boost their income. One of the cornerstones of illegal and unethical puppy trading is prioritising profitability, commonly at the expense of welfare [15]. A study documenting the movement of farmed puppies bred in rural areas of Wales (Wales being geographically 85% ‘rural’ [15]) and sold on urban markets supports the theory that puppies are being bred in Wales to be sold elsewhere [24]. The density of advertisements in Wales could suggest that a large part of the online puppy trade is being produced to meet demand in other areas of the UK, and, thus, a large number of puppies are being moved across country borders (by sellers or new owners), potentially undergoing long journeys with unknown welfare risks. Further investigation into this would be beneficial, if puppies are being transported long distances, this could suggest the trade via a third party, in violation of the law against third party sales.

A decrease in advertisements between phases (~−1700 in total) could have resulted from the COVID-19 Lockdowns that began in March 2020 [41] and subsequent travel restrictions, both within country and across international borders [42]. That being said, the RSPCA estimated that 29% of puppies were imported for sale on the UK market [2], but only a −12.48% decrease in advertisement counts was noted between Phases 1 and 2 [18,43]. This discrepancy could be due to recent indications from NGOs that puppies continued to be imported during 2020–2021, and supports the theory that importers having circumvented conventional and COVID-19 restricted travel pathways, or due to a general increase in puppies sales reported during the COVID-19 lockdowns [44].

A decrease in advertisements between phases may challenge the acute increase in demand for puppies noted over the COVID-19 Lockdown [45]; however, it could be that supply may have been unable to meet this acute increase in demand immediately due to the ‘production lag’ of canine gestation (between 58 and 68 days) and weaning (ideally at least 56 days to completion [46]).

With regard to websites hosting puppy advertisements, Pets4Homes had the highest volume of advertisements for puppies (57.21% of all advertisements), perhaps reflecting its animal-specific, rather than general, focus. The use of Pets4Homes increased between the two phases. All three sources are members of the Pet Advertising Advisory Group (PAAG), an advisory group consisting of stakeholders working against irresponsible advertising of pets for sale, and have voluntarily committed to adhere to their minimum standards for advertising puppies [47].

Our final objective was to assess breed frequency overall and, more specifically, frequency of breeds linked to inherited disease/conformational disorder. A large majority (66%) of advertisements advertised only 20 breeds, reinforcing breed desirability/fashion as a significant factor in the market. Almost half of the dogs advertised were breeds identified in Asher et al.’s [3] review of the KC’s 50 most popular breeds and their links to conformational disorder (CD). This implies that dogs with CD are extremely popular. It could thus be hypothesised that the popularity of these breeds has motivated an increase in production and breeding for specific traits, despite the related welfare implications [48].

Interestingly, although frequency of advertisements for both CD and non-CD dog advertisement frequency increased between phases, non-CD dogs experienced a larger increase. This suggests that non-CD breeds were catching up in popularity and, in fact, an 8.37% decrease in advertisement counts advertising brachycephalic breeds was observed in Phase 2; perhaps recent efforts from stakeholders to educate the public on the welfare issues associated with these breeds has led to a decrease in their popularity [49,50,51]. It could also be suggested that the public’s motivation for puppy buying, and thus breed preference, changed in Phase 2 due to the COVID-19 Lockdown stay-at-home order and associated restrictions [41]. Indeed, a Royal Veterinary College (RVC) report suggested that ‘exercising more’ was one of the top three reasons people bought a dog during the pandemic [45]. As such, due to possible respiratory disorders, brachycephalic dogs may not be a first choice [52].

With regard to price, the mean advertisement selling price was ~GBP 927.14, reportedly high for a market on which payment is almost entirely made in cash [53]. Cash markets disable accountability, especially as many puppies are bought without receipts [24], severing contact post-purchase. At the 2018 British Small Animal Veterinary Association (BSAVA) conference, Scottish Society for the Prevention of Cruelty to Animals (SSPCA) Officer Mark Rafferty suggested that consumers’ willingness to pay such large sums of cash was an interesting insight into high-risk consumer behaviour and the fervent desire to purchase puppies [53]. This supports the theory that consumers may be acting ‘irrationally’ due to the reported ‘marketable emotional value of puppies’ [54,55]. He also reports consumers being pressured or threatened by sellers [53]. This supports a theory that crime syndicates are using the puppy trade as a way to launder money made from other illicit activity [19]. An increase in prices between phases (+GBP 243.58) could reflect the reported ‘pandemic puppy’ demand, outweighing existing supply, with 12.48% fewer advertisements posted in Phase 2 compared to Phase 1. As in other product markets, price is the result of shifts in supply relative to demand [56]. Not only did demand increase acutely after March 2020, if we assume that supply was limited based on lower advertisement count (e.g., due to production lag, travel restrictions), this could have induced a ‘perfect storm’ resulting in price inflation. This trend would need to be observed for a longer period of time to prove meaningful. The most expensive breed on average (Bulldog) was only the 7th most advertised breed, possibly due to the more complicated process and, thus, higher ‘production cost’ involved in breeding bulldogs, whose extreme conformation may require aided-mating, artificial insemination, and c-section to facilitate pure-breeding [57]. The French Bulldog is the only popular breed whose price fell in Phase 2 (−2.7%), all other popular breed’s prices increased substantially. Perhaps this is because the French Bulldog has been central to many campaigns warning consumers away from brachycephalic breeds, due to its reported extreme popularity over the last decade [58]. It could also be in conjunction with the suggested higher demand for more ‘active’ breeds represented by higher advertisement counts for breeds not linked to CD. This is potentially supported by the ‘catch-up’ effect in price across phases for non-CD breeds, described above.

Over both phases, 46.85% of advertisements were for breeds linked to CD despite making up only 3.4% of the unique set of breeds in the data. This may actually be an underestimation due to the non-inclusion of popular designer cross breeds in the definition of CD used here, with Asher et al. [3,9] using only the top 50 most registered breeds in the KC register for their study. Although, in theory, ‘regulated’, ‘legal’, and ‘responsible’ breeders practice ‘safe inbreeding’, outlined by the KC, it is still a practice that risks dog welfare. The KC themselves note at the top of each KC breed standard reference sheet that:

“Breeders and judges should at all times be careful to avoid obvious conditions or exaggerations which would be detrimental in any way to the health, welfare or soundness of this breed” [59].

Despite this, various breed standards include descriptions of conformations with obvious, proven, and often severe links to disease and disorder, including “long body”, “legs short but strong in bone”, “Ears long; reaching only slightly beyond end of muzzle”, and “skin on and about head slightly loose and finely wrinkled”; these are often followed by vague phrases such as “but not excessively” [59]. It could be argued that dogs being bred with conformations that lead to dwarfism, respiratory disorders, skin fold dermatitis, otitis, and spinal disease (amongst others) is the knowing creation of animals that cannot possibly benefit from the Five Welfare Needs of Animals under the Animal Welfare Act (2006), most notably freedom from pain and disease (not only due to CD, but to the treatments required to rectify them) and freedom to enjoy natural behaviours (due to malformation of joints, respiratory issues, or risk of spinal injury) [60,61]. In addition, it is in breach of the LAIA (England) regulations, wherein no dog may be kept for breeding if it can be reasonably expected that breeding from it could have a detrimental effect on its or its offspring’s health or welfare. It could thus be suggested that the creation of dogs to these breed standards is an inherent breach of these law and, as such, should not be considered legal.

If patterns in puppy advertisements are a reflection of consumer behaviour as in other industries [62], the findings of this study suggests that consumers are revealed to be choosing to purchase dogs that may have been farmed and/or bred to potentially suffer. The so called ‘baby-schema’, a preference for ‘baby-faced’ dogs, however, may be encouraging the decline as per these data [10]. Further study into market trends and the relationship between breed preference, and intrinsic ownership style and breed trends in relinquishment would be interesting future investigations. Using the methodology of this study, research to measure changes in advertisement frequency for breeds after CD-specific, targeted intervention, or perhaps tailored to specific ownership styles could also be helpful for more effective campaign planning. A linguistic study into the wording used in puppy advertisements could explore the theory of marketing language as a reflection of consumer behaviour, for example the use of aesthetic vs. health descriptors in advertisement titles [62]. If successful, these studies could go some way to support legislation that holds consumers accountable for their puppy purchasing behaviour.

## 5. Conclusions

A 2017 scoping review on the UK sourcing of puppies concluded that ‘developments in online trade exacerbate the problems inherent in the puppy industry, including negative animal and human health, economic and environmental consequences’ [15]. In 2016, less than 5% of puppies sold in the UK had originated in registered pet shops, with the majority being purchased online from classified advertisements [2]. The internet has now been recognised as one of the main enablers for irresponsible and illegal selling and buying behaviours and, similarly to online technology, the UK puppy trade has outpaced existing regulation [2,15,29,63]. The anonymity and the opportunity to reduce face-to-face contact with consumers, makes online advertisement the ideal sales platform for breeders who wish to conceal their facilities and identity [19]. Additionally, these supply data reveal sellers as favouring ‘fashionable’ dog breeds, which reflects a faithfulness to impulsive and intrinsic consumer demand. As a possible consequence, more dogs and puppies may be being bred to suffer short and long-term welfare consequences, including disease, deformity, and risk of abandonment [2].

This study demonstrates the use of web-scraped data, along with non-invasive and inexpensive methods, to provide an evidence base with which to support intervention or legislative reform. For example, further research could allow for the quantification of lost HMRC revue due to reported tax evasion from online puppy sales, potentially bolstering governmental intention to enforce regulations against puppy traders. A tool to link sellers into working networks, based on repeat usernames, phone numbers, or stock images, could provide an effective method for exploring puppy transportation within and between country borders, and adherence to breeding legislation (i.e., requirement of breeding licensing [64]).

Our findings also support assumptions regarding the existence of a consumer culture, largely motivated by trends and intrinsic behaviour, with one of the most concerning findings including the overall favouring of breeds linked to conformational disorder. As per Maher and Wyatt (2019) [24], the neo-liberal character of this market is enabling the breeding and purchasing of beings bred to suffer. This is evident not only in the consistent popularity of dogs with known conformation or inherited disorders, but also in the sheer number of advertised puppies on these websites, a substantial number of which may face abandonment, considering that 20% of dogs are reportedly abandoned within 2 years of purchase [65]. This illustrates a clear need for consumer and owner regulation that is based on accountability.

Recommendations for further study include the continued development of the methodology of this paper; analysis of how the data are affected by legislation coming into force; study of micro-patterns (e.g., date, breed, and precise location); the automation of more routine and more extensive web scraping; the exploration of social media markets, as well as bolstering this information with other datasets, including Local Authority enforcement records, import records, and veterinary archives. Together, these represent an excellent and impactful future for this area of research.

## Figures and Tables

**Figure 1 animals-13-01682-f001:**
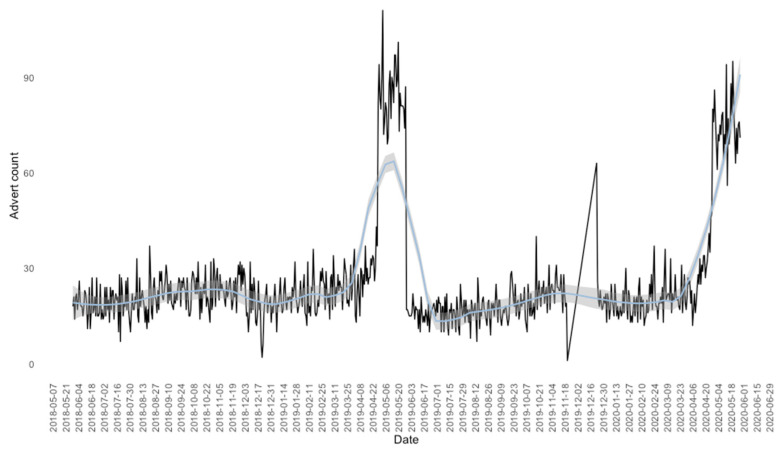
Advertisement frequency over the study period. Total advertisements per day, from 1 June 2018 to 31 May 2020 (i.e., a 24-month period) acquired from 3 classified advertisement platforms. Variation in number of advertisements per day highlighted by solid black line, overlaid by smooth LOESS fit (light blue). Seasonality was observed in both phases (*p* = 0.38, Augmented Dickey–Fuller), with advertisement frequency peaking in April and May. Several weeks were unaccounted for in December and May of Phase 2, thus these months were discounted from the downstream analysis.

**Figure 2 animals-13-01682-f002:**
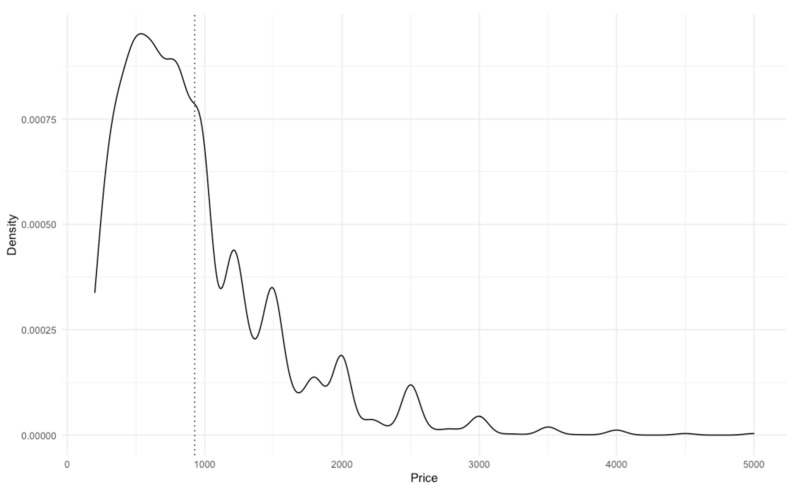
Density of advertised price. The mean advertisement price was GBP 927.14 (SD = GBP 604.03). Additional peaks were observed at notable values, e.g., GBP 1250, GBP 1500, GBP 2000, GBP 2500, etc.

**Figure 3 animals-13-01682-f003:**
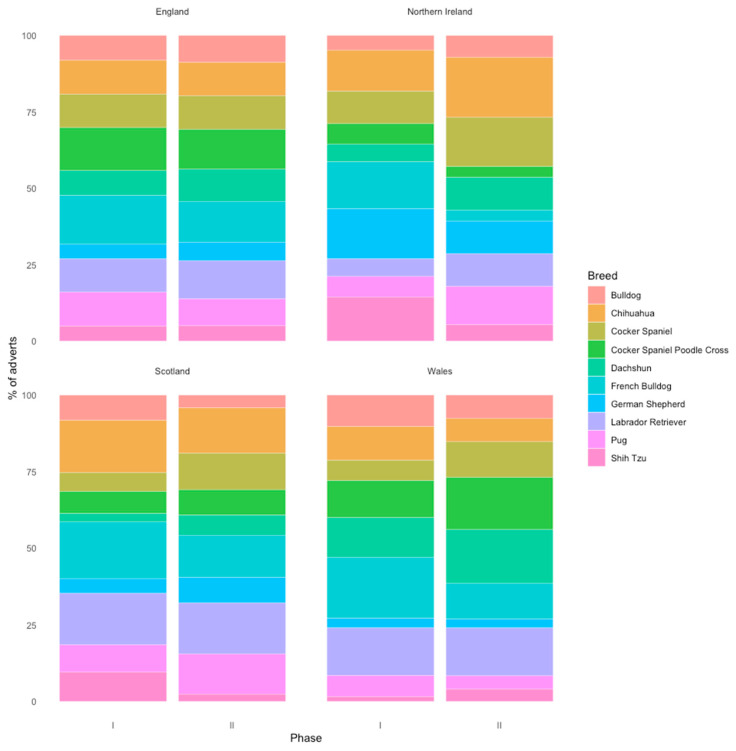
Breed frequency by region. Stacked bar charts representing percentage of advertisements per region, posting top 10 most popular breeds overall.

**Figure 4 animals-13-01682-f004:**
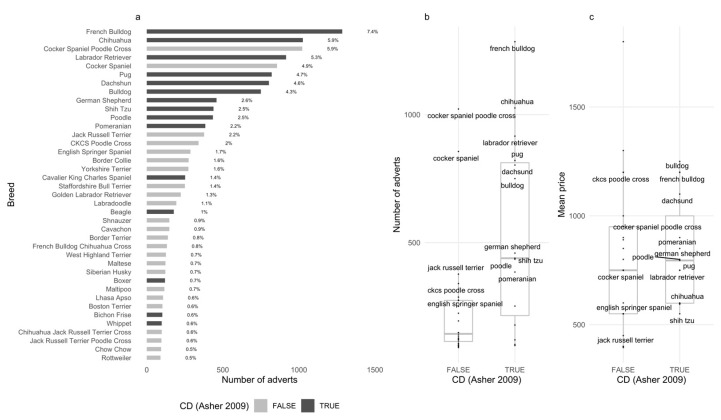
Breeds linked with conformational disorders are advertised more frequently and at a higher price. (**a**) Total number of advertisements posted per breed, across full study period. Breeds representing <0.5% of total advertisements were removed. Posts advertising breeds linked with conformational disorders are highlighted (black) and make up over (46.85%) of all advertisements. (**b**) Boxplot representing the frequency of advertisements for breeds linked to conformational disorder (TRUE), compared to those without (FALSE)t. Breeds linked to CD were more commonly advertised than those without (β = 298.64, SE = 96.74, 96.74, *p* < 0.001). Here, a positive β value of 298.64 represented the mean difference in advertisement counts between non-CD (mean = 239.29, SD = 232.42) and CD breeds (mean = 537.93, SD = 373.61), which is more than double. (**c**) Boxplot representing frequency of mean price per breed. Though CD breeds (TRUE) were observed to be more expensive than non-CD breeds (FALSE) on average, this trend was found non-significant when assessed across all breeds (β = 31.96, SE = 100.07, *p* = 0.75) [3].

**Figure 5 animals-13-01682-f005:**
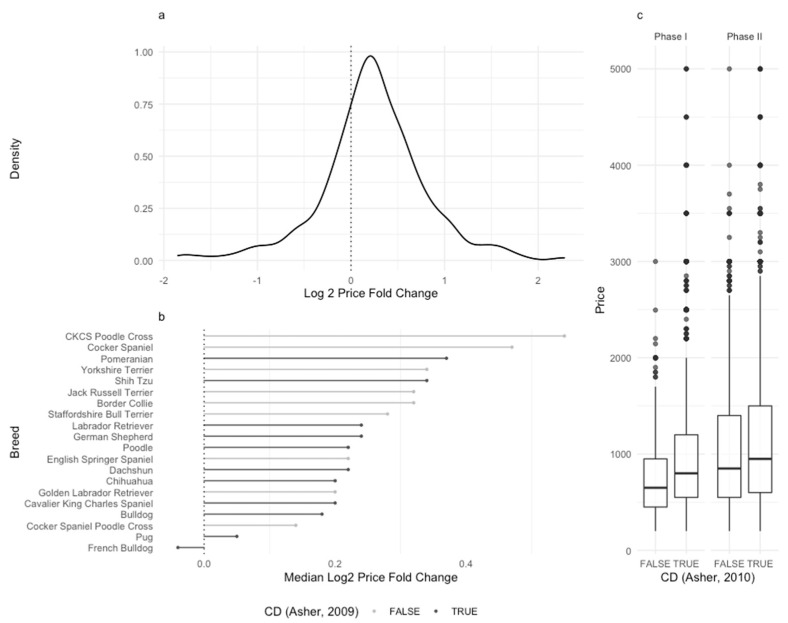
Price increases between phases. A per-breed price fold change was calculated as the Phase 2 price as a proportion of its corresponding Phase 1 value. (**a**) The distribution of log_2_ advertised price fold change (*x*-axis) demonstrated an increase in Phase 2 prices (mean = 0.24, 95% CI = [0.16, 0.31]). (**b**) Log_2_ price fold change (*x*-axis) per breed (*y*-axis) for the top 20 most popular breeds were uniformly increased in Phase 2, except for French Bulldogs, which decreased. (**c**) The distribution of advertised price (*y*-axis) split by Phase and CD status (*x*-axis, where True is CD and False is non-CD). The increase in price between phases was most pronounced for non-CD breeds (*p* < 0.001) [3,9].

**Table 1 animals-13-01682-t001:** Data were organized into two 365-day phases.

Phase	Start	Finish
**Phase 1**	1 June 2018	31 May 2019
**Phase 2**	1 June 2019	31 May 2020

## Data Availability

The datasets presented in this article are not readily available due to existing data sharing agreement between Dogs Trust (registered charity number 227523 in England and SC037843 in Scotland) and Hindesight Ltd. (Registered 08216374, UK). Requests to access the datasets should be directed to kirsten.mcmillan@dogstrust.org.uk.

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
