# Peer review of "What Patterns in Online Classified Puppy Advertisements Can Tell Us about the Current UK Puppy Trade"

_animals, 2023, doi:10.3390/ani13101682_

Round 1
Reviewer 1 Report
Thanks for an interesting read. I enjoyed your manuscript and commend you on a substantial effort to understand your goals.
L12, 17 & 65 (possibly elsewhere): Make sure that acronyms are explained in full at first use.
L25: I assume it is 46.85% of all puppies in the sample (not just the 66%). If so add "all", if not please specify.
L26 + 49 et al.: conformational disorders (again perhaps pedantic, but I would say that or "disorders of conformation")
L26 and throughout: use "advertisement(s)" rather than "advert(s)" (perhaps a pedantic point).
L31: I note that the scientific abstract is in a format similar to that used by veterinary journals. I assume this is OK within the animals formatting.
L33: I would suggest "ethically concerning" rather than "unethical" which is IMO rather too definitive.
L35: replace "both" with "some" (indicating this is pervasive but not absolute)
L38: explain the term "webscraped"
L39: incomplete sentence
L44: "were restricted" should be at the end of the sentence.
L47: How was this P-value derived (and what are the test stats if relevant)? The methods section of the abstract should provide some indication as to the stats used.
Given the detailed results you have provided I think the abstract should contain more of the key findings.
L64: I'm not sure that the capitalist structure of pet sales is contrary to values of the HAB. In fact, for some, the monetary value of the dog may be an intrinsic part of the breed selection process and hence integral to the HAB/R. I do agree with L65-68.
L71: most breeders would say "line breeding" (ie the breeding of closely related individuals but not siblings/parents). I know this is clearly inbreeding.. but it is perhaps worth using the industry terms.
L78-9: delete "in puppies breed in Britain" (you have already said "British-bred"). The words "to good standards" seem to be floating here, they don't seem linked to the main thrust of the sentence. I think there is a need to rewrite the sentence to include "... a deficit of British puppies bred to a high standard..."
L77: UK should be described in full before this point so UK is fine.
L81: is it British or UK? (It depends on whether any of the refs included N. Ireland in which case it is "British and UK")
L102: "Anonymity and the opportunity..."
L110-113: I would divide this into two sentences so you can better explain the term "cost gap" (I assume you mean that unscupulous breeders can ask lower prices?)
L110-116: I feel these points could be made more clearly/succinctly
e.g. "Online systems distort the market for puppies by providing unscrupulous breeders with a direct tool with which to financially undercut those breeding to high welfare standards. Furthermore a lack of buyer screening and direct delivery of puppies can increase temptations to purchase. This is due to increased consumer convenience and immediate availability which run contrary to good breeding/sale practices."
L121-125: break the sentence. Remove "whilst" from L121 and add "...(18). However, they do not..." in L124
L129 add "human" or "consumer" to behavioural intervention
L135: again, explain the "webscraping" process. What words were looked for? Were there inclusion/exclusion criteria (e.g. did the search exclude things like "free to a good home" etc.)? It is useful for the reader to have more info on how data were identified for scraping, especially if they wish to replicate the process.
How were Xbreeds defined in terms of CD? Were they assumed to not have any? If so, why?
L171: what are the numbers in parentheses?
L174: You need to explain how this did not comply with HERC? Are you suggesting it did not comply with their GDPR requirements?
L183: relative difference of what?
L186: what are fold changes? (excuse my ignorance)
L214: Put "were restricted" at the end of the sentence
Table 1: I don't think this is necessary given it is provided in text
L258: 48% of ads
L289: disorders not deformity
Fig 4&5: can you explain what "false/true" mean? If you mean with/without CD I would suggest this is a better description.
L328: explain "close the gap" (ie be specific as to what you mean)
L365: is dog ownership per capita higher in Wales? This could be an alternative explanation as to why more dogs are on the Welsh market.
L376: I think this is perhaps an over reach from your data given there is no info about on-selling or transaction.
L406: delete "of these breeds"
L458-472: I accept this is a fair assessment of the situation, however it seems unrelated to the core of your objectives and results. As such, I would remove it from the discussion. You may want to add a couple of short sentences to the intro around CDs and their welfare implications.
Author Response
Many thanks for your inciteful, constructive and well thought out comments. We think the manuscript has been much improved as a result! Please see below for a detailed description of changes in response to each of your comments.
Please let us know if you have any further questions or comments.
Kind Regards.

Reviewer 2 Report
Good paper with only few comments.
Line 39: it is missing the end of the phrase
Line 49: give some examples of conformation disorders… like…
Line 91: give some examples of chronic inherited conditions…
Line 146: explain how the same add in different websites was treated to avoid doubles…
Line 176: Is Appendix D really necessary?…
Line 359: induced oestrus in bitches is not a common practice. The protocol is good for oestrus induction but is not great for ovulation. The sharp increase in April-May could be explained by a dormitory effect (Concannon, 1993) or a different source of puppies to supply the demand.
Line 390 – ideally, at least 56 days.
Line 329 – the age of the owner seems to be a confounding effect as explained in the discussion.
Line 471 – welfare (.
Line 471 – suggest)
Author Response

(The authors gave the same response as above.)
